# Ovulation in *Drosophila* is controlled by secretory cells of the female reproductive tract

Jianjun Sun[1,2], Allan C Spradling[1,2]*

[1]Department of Embryology, Carnegie Institution for Science, Baltimore, United States; [2]Howard Hughes Medical Institute, Baltimore, United States

**Abstract** How oocytes are transferred into an oviduct with a receptive environment remains poorly known. We found that glands of the *Drosophila* female reproductive tract, spermathecae and/or parovaria, are required for ovulation and to promote sperm storage. Reducing total secretory cell number by interfering with Notch signaling during development blocked ovulation. Knocking down expression after adult eclosion of the nuclear hormone receptor Hr39, a master regulator of gland development, slowed ovulation and blocked sperm storage. However, ovulation (but not sperm storage) continued when only canonical protein secretion was compromised in adult glands. Our results imply that proteins secreted during adulthood by the canonical secretory pathway from female reproductive glands are needed to store sperm, while a non-canonical glandular secretion stimulates ovulation. Our results suggest that the reproductive tract signals to the ovary using glandular secretions, and that this pathway has been conserved during evolution.

## Introduction

The oviduct must interact extensively with the ovary to receive ovulated eggs in a manner that maximizes successful reproduction and minimizes egg loss and ectopic pregnancy. In humans, oviduct-ovary signaling also likely influence serous ovarian carcinoma development, a disease now thought to originate from the secretory epithelia of the distal oviduct (Fallopian tube) following an extended number of ovulatory cycles (*Levanon et al., 2008*; *Bowtell, 2010*; *Kurman and Shih Ie, 2010*; *King et al., 2011*; *Berns and Bowtell, 2012*). How the oviduct normally influences mammalian ovulation remains unclear, although a great deal has been learned about the hormonal control of ovulation within the ovary itself (*Kim et al., 2009*; *Conti et al., 2012*), and genes such as the nuclear hormone receptor *Lrh-1* are known to be essential (*Duggavathi et al., 2008*). The growing realization that important aspects of gamete biology have been conserved during evolution suggests that insights into oviduct-ovary signaling may come from studies of model systems.

The *Drosophila* oviduct plays important roles during egg production that may involve communication with the ovary. The oviduct must be prepared to transport each oocyte released from the ovary to the uterus, to mediate its water uptake and eggshell crosslinking, and to position it for efficient fertilization (reviewed in *Spradling, 1993*). During each cycle of ovulation, just one of the many mature oocytes present in the two ovaries is released into an oviduct. Octopaminergic neurons innervating oviduct muscle and epithelia are needed for ovulation, probably to activate oviduct muscle contraction and to stimulate epithelial secretion by activating the Oamb octopamine receptor (*Lee et al., 2003*, *2009*; *Monastirioti, 2003*). The steroid hormone ecdysone is produced in the adult ovary and is required to maintain egg production (*Buszczak et al., 1999*), although a specific role in ovulation has not been tested.

Glandular secretions from male reproductive tracts in both invertebrates and vertebrates facilitate reproduction at multiple steps (*Bloch Qazi et al., 2003*; *Suarez, 2008*; *Heifetz and Rivlin, 2010*;

*For correspondence: spradling@ciwemb.edu

Competing interests: The authors declare that no competing interests exist.

**eLife digest** Mammalian oviducts, or Fallopian tubes, convey egg cells from the ovaries to the uterus. Signalling between the ovary and oviduct, and secretory products produced throughout the reproductive tract, help to increase the likelihood of conception, minimise the loss of egg cells, and reduce the risk of ectopic pregnancy (in which an embryo implants outside the uterus). These processes may also influence the development of ovarian cancer, since Fallopian tube secretory cells were recently identified as the source of the most common and lethal subtype of epithelial ovarian cancer, high grade serous ovarian cancer.

Oviduct to ovary signalling is poorly understood in mammals. However, experiments using model organisms such as the fruit fly (*Drosophila melanogaster*) provide a potentially powerful approach to the problem, since many mechanisms in gametogenesis are conserved between species. In particular, secretions within the *Drosophila* female reproductive tract appear to boost reproductive success by interacting with sperm cells and seminal proteins, as in mammals. But whether these secretions reach the ovary and influence ovulation, or simply act on other aspects of reproduction such as mating, sperm storage, fertilisation or egg laying, remained unknown.

In this study, Sun and Spradling identified new genes controlling reproductive gland development and used this knowledge to elucidate secretory cell function. By mutating these genes, or the nuclear hormone receptor *Hr39*, they were able to reduce the total number of secretory cells that developed in the female reproductive tract, or to alter their function in adults. The ovaries of flies with abnormal secretory cell function contained as many egg cells as those of normal flies, but the mutant females laid fewer eggs. This indicates that secretory cells are required for at least one stage of reproduction.

By comparing ovulation rates in mutant and normal flies, Sun and Spradling showed that the secretory cells generate a product that is specifically required for ovulation, and that production depends on Hr39 activity. This Hr39-dependent secretion is a good candidate for a conserved signal between the reproductive tract and ovary because mouse *Lrh-1*, a mammalian gene closely related to *Hr39*, is expressed in oviduct secretory cells and is itself required for ovulation. The secretory cells were also found to produce protein secretions that are necessary for female flies to store sperm in the reproductive tract after mating.

By elucidating the roles played by female reproductive tract secretions, and demonstrating that they include a signal to the ovary that stimulates ovulation, the work of Sun and Spradling may lead to an increased understanding of ovarian cancer in humans.

*Holt and Fazeli, 2010*; *Ikawa et al., 2010*; *Jeong et al., 2010*; *Manier et al., 2010*; *Dunlap et al., 2011*). Multiple proteins produced in the *Drosophila* male accessory glands are mixed with sperm upon ejaculation and transferred to the female reproductive tract where they mediate sperm storage, capacitation, and maternal reproductive behavior (*Avila et al., 2011*). For example, sex peptide (SP) increases egg laying and reduces female receptivity (*Chen et al., 1988*; *Chapman et al., 2003*; *Liu and Kubli, 2003*) by binding to a specific receptor, SPR, in three sets of $fru^+ppk^+$ sensory neurons in the female reproductive tract (*Yapici et al., 2008*; *Hasemeyer et al., 2009*; *Yang et al., 2009*). Ovulin, a protein transferred in male seminal fluid induces ovulation shortly after copulation (*Herndon and Wolfner, 1995*; *Heifetz et al., 2005*). Another transferred peptide, Acp36DE, facilitates sperm storage (*Neubaum and Wolfner, 1999a*; *Avila and Wolfner, 2009*). Ejaculate components produced in the mammalian testis, prostate, and epididymis also play important roles in reproduction (*Suarez, 2008*). For example, mammalian spermadhesins secreted from seminal vesicles mediate sperm attachment to the oviduct epithelia (*Talevi and Gualtieri, 2010*).

Female reproductive tract secretions also boost reproduction by interacting with transferred sperm and seminal proteins in many species (*Holt and Fazeli, 2010*; *Jeong et al., 2010*; *Dunlap et al., 2011*; *Franco et al., 2011*; *Schnakenberg et al., 2011*; *Wolfner, 2011*). *Drosophila* spermathecae and parovaria, the major exocrine glands of the female reproductive tract, are required for fertility and sperm storage (*Anderson, 1945*; *Allen and Spradling, 2008*; *Schnakenberg et al., 2011*). Whether *Drosophila* female secretory products regulate other aspects of reproduction remains poorly understood, however. Recently, reproductive secretory cell development in the spermathecae and parovaria

was shown to follow a stereotyped cell lineage and to depend on the transcription factor Lozenge (*Anderson, 1945*) and Hr39 (*Allen and Spradling, 2008*; *Sun and Spradling, 2012*), a nuclear hormone receptor homologous to Lrh-1.

Here we used our new understanding of reproductive gland development to manipulate the number and activity of secretory cells in adult females. In addition to documenting a role for protein secretion in sperm storage, we show that adult Hr39 expression and a non-canonical secretion from the adult female reproductive glands are required for normal ovulation. Thus, ovulation in both *Drosophila* and mice depends on the homologous nuclear hormone receptors Hr39 and Lrh-1. Our results suggest that a conserved program of reproductive tract secretion mediates oviduct-ovary signaling and may be relevant to the origin of ovarian cancer.

## Results

### Female reproductive glands are essential for ovulation

The overall function of female reproductive glands can be assessed by studying adult females bearing mutations in *lz* or *Hr39* which disrupt their development (*Anderson, 1945*; *Allen and Spradling, 2008*). Mutants retain only rudimentary glands or lack them entirely and show defects in sperm storage and egg laying (*Anderson, 1945*; *Allen and Spradling, 2008*). All $lz^{-/-}$ females completely lack reproductive glands, while $Hr39^{-/-}$ females either lack glands (>90%) or retain a single defective spermathecae with very few secretory cells (*Figure 1A–C*). The ovaries in *Hr39* and *lz* mutant females contain a full complement of mature oocytes, however, both lay significantly fewer eggs than controls (*Figure 1D–E*), indicating that secretory products are required for one or more steps downstream from oocyte completion, such as ovulation, mating, sperm storage, fertilization, or egg laying.

It would be particularly interesting if reproductive glands were required for ovulation, since this might indicate that secretory products coordinate activities of the ovary and reproductive tract. Consequently, to distinguish whether ovulation was specifically affected, individual female flies were examined to determine if oocytes had left the ovary within 6 hr after mating to wild type males. In controls, about 50% of females initiated ovulation within this time interval, as indicated by the presence of an egg in either the oviduct, the uterus, or the food vial (*Figure 1F,G*). In contrast, none of the *Hr39* mutant and only 2% of *lz* mutant females initiated ovulation. The failure of mutant animals to ovulate was not due to defects in mating, as both mutant and control females showed similar rates of copulation success (indicated by the presence of sperm in the female reproductive tract) (*Figure 1H,I*). Thus both *Hr39* and *lz* are required for ovulation, at least initially.

### *lz* and *Hr39* are not required in reproductive tract neurons

Before determining which cells within the glands were needed for ovulation, we investigated whether the failure of *lz* and *Hr39* mutant females to ovulate was due to a secondary requirement of these genes outside of the reproductive glands. For example, *lz* and *Hr39* might function in the octopaminergic neurons that innervate oviduct muscle and stimulate oviduct epithelial cells prior to ovulation (*Lee et al., 2003*), or in the $fru^+ppk^+$ sensory neurons of the female reproductive tract. However, *lz* expression could not be detected in the octopaminergic neurons innervating the oviduct nor in the oviduct muscle or epithelial cells (*Figure 2A*). Lineage tracing also showed that oviduct cells are not derived from $lz^+$ cells (*Sun and Spradling, 2012*). Yet blocking the proliferation of *lz*-expressing cells during pupal development was sufficient to disturb ovulation (*Table 1*).

Further experiments argued against a requirement of *lz* and *Hr39* in the $fru^+ppk^+$ sensory neurons for the female postmating behaviors elicited by the SP/SPR signaling pathway (*Hasemeyer et al., 2009*; *Yang et al., 2009*). *lz* was found to be expressed in a subset of these neurons near the oviduct-uterus junction, but $fru^+ppk^+$ neuron number was not affected by knocking down *cycA* or *Hr39* using the *lzGal4* driver (*Figure 2A,B*). Likewise, *Hr39* does not function in these neurons because no defects were observed in egg laying when Hr39 levels were reduced by driving $Hr39^{RNAi}$ expression using *ppk-Gal4* (*Figure 2C*). Furthermore, ectopically expressing a membrane-attached form of sex peptide (mSP) in $fru^+$ neurons blocked virgin female receptivity even when carried out in an *Hr39* mutant background (*Figure 2D*) (*Hasemeyer et al., 2009*; *Yang et al., 2009*), indicating that *Hr39* mutant females have intact neural circuitry for post-mating behavior. Yet these same females still did not lay eggs (*Figure 2E*). Thus, the disruption in ovulation observed in *lz* and *Hr39* mutant females is unlikely to be caused by altered SP/SPR signaling or to other neural defects within the reproductive tract.

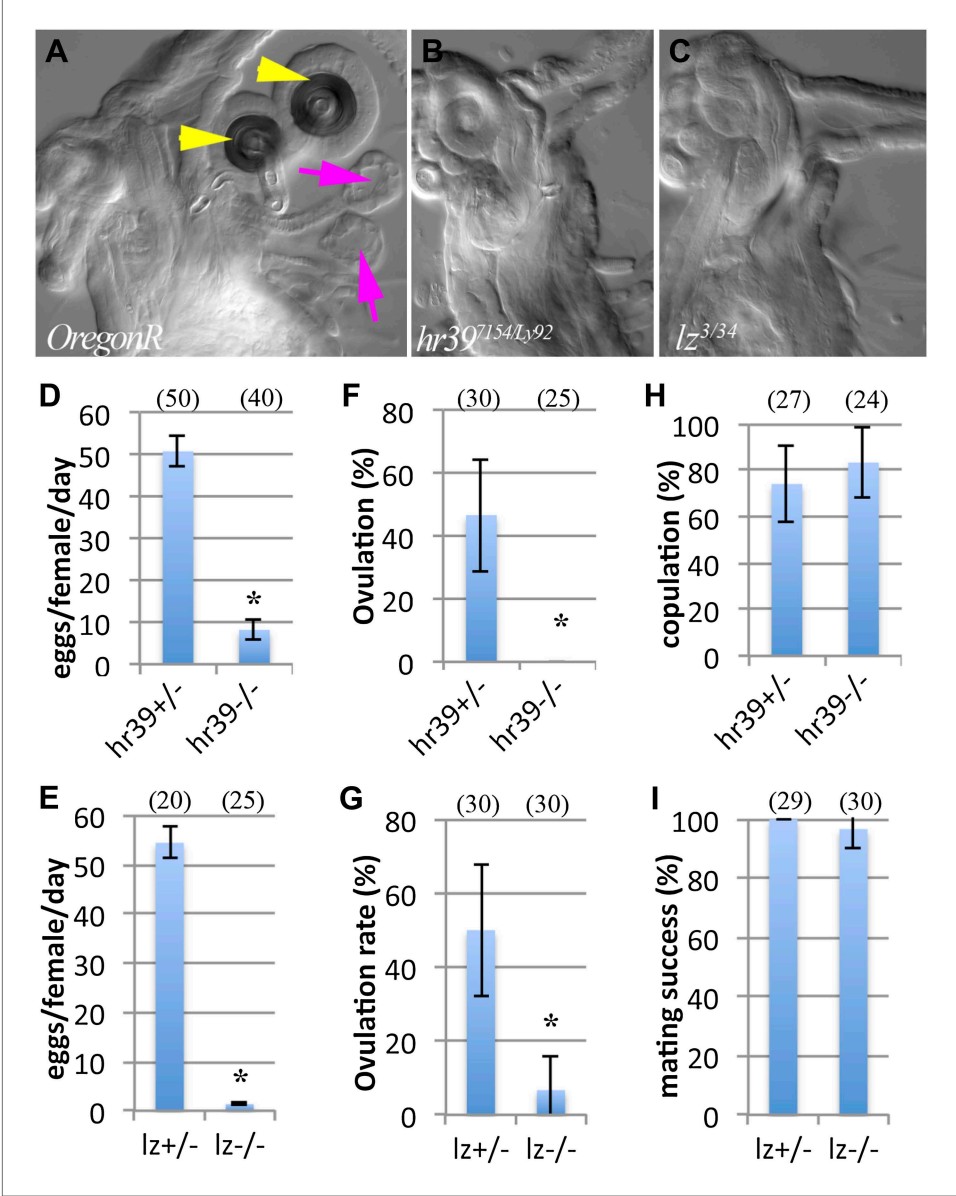

**Figure 1**. Female reproductive glands are essential for ovulation. (**A**)–(**C**) DIC images of *Oregon-R* (**A**), *Hr39$^{7154/Ly92}$* (**B**) and *lz$^{3/34}$* (**C**) mutant female lower reproductive tracts. Both spermathecae (yellow arrowheads) and parovaria (magenta arrows) are absent in the mutant animals. Bar graphs display the rate of egg laying (**D** and **E**), ovulation frequency (**F** and **G**), and copulation frequency (**H** and **I**) for the two mutant genotypes, and heterozygous controls. In all figures, the number of egg laying groups or mating pairs is shown in brackets. Error bars are SEM, or 95% confidence intervals. *p<0.01 (Fisher's exact test, or Student t-test).

## Notch signaling and Hindsight are required to form reproductive gland secretory cells

In order to analyze which cells within the reproductive glands are required for ovulation, we developed methods for perturbing gland development more precisely than is possible using *lz* and *Hr39* mutations. Several previous observations during studies of pupal spermathecal and parovarial development (**Sun and Spradling, 2012**) revealed a likely role for Notch signaling in their developing secretory cells (SCs). The stereotyped divisions of secretory unity precursor cells (SUPs) resemble the Notch-requiring divisions during peripheral nervous system development (**Lai and Orgogozo, 2004**; **Sun and Spradling, 2012**). Consistent with this idea, we discovered that a Notch signaling reporter is

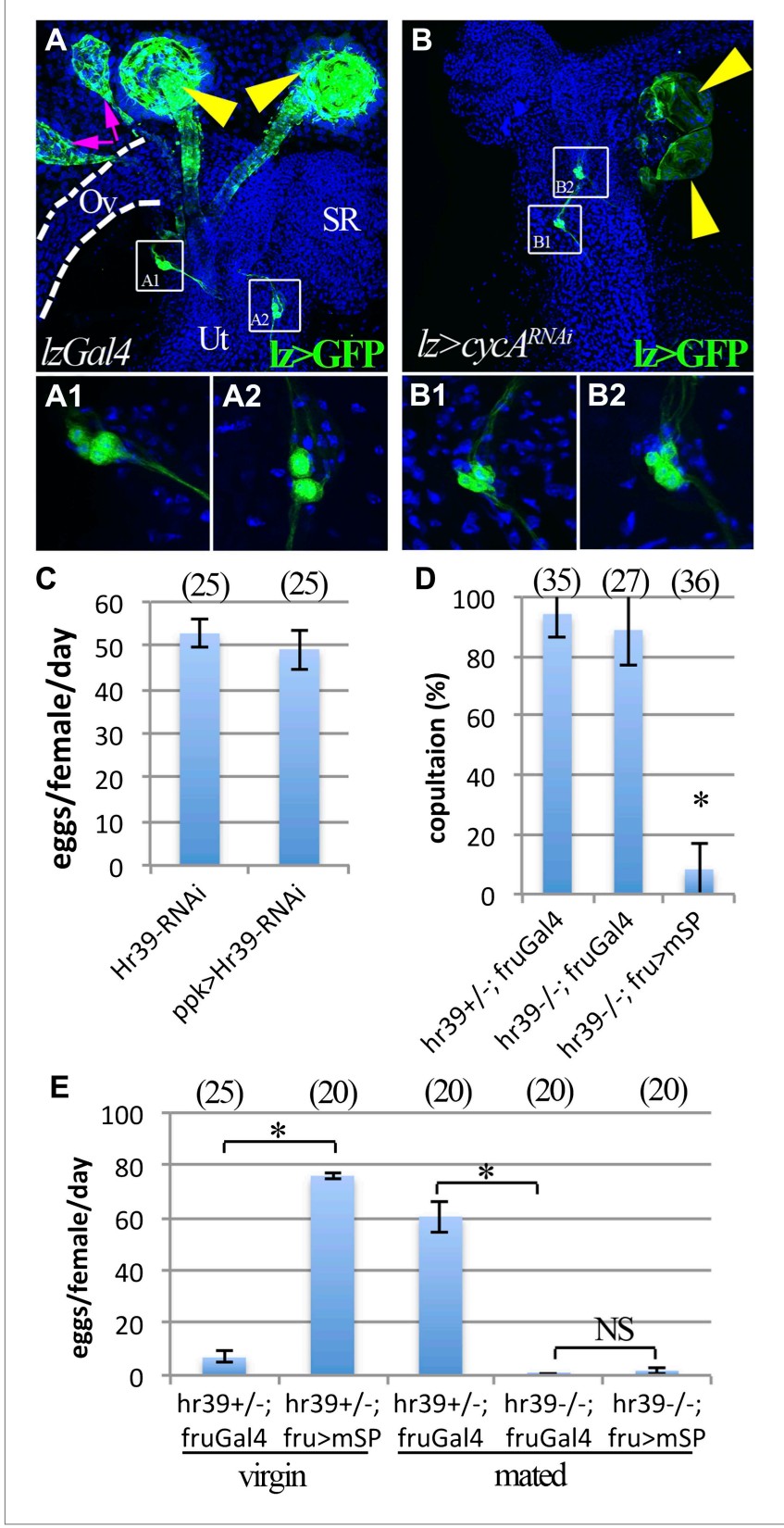

**Figure 2**. *lz* and *Hr39* are not required in reproductive tract neurons. (**A**) *lz* expression (*lzGal4* driving UAS-mCD8::GFP) in control female reproductive tract. Spermathecae (yellow arrowheads); parovaria (magenta arrowheads). *Figure 2. Continued on next page*

*Figure 2. Continued*

Ov: Ovuduct; SR: Seminal receptacle; Ut: Uterus. Two sets of *lz*[+] sensory neurons are illustrated at higher magnification in (**A1** and **A2**). (**B**) *lz* expression in female reproductive tract expressing *lzGal4>UAScycA* (*lz>cycA*[RNAi]). *lz*[+] sensory neurons are not affected (**B1** and **B2**). (**C**) Egg production is not affected by expressing *Hr39-RNAi* in *ppk*[+] neurons of the reproductive tract. (**D**) Ectopic expression of mSP in *fru*[+] reproductive tract neurons reduces virgin female copulation rate, even when neurons are mutant for *Hr39*. (**E**) Ectopic mSP in *fru*[+] neurons is sufficient to induce egg laying in control virgin females but not in *Hr39*[−/−] females even in the presence of males. * indicates p<0.01 and NS indicates p>0.05.

dynamically expressed in this lineage (*Figure 3A–D*, green). Hindsight (Hnt), a transcription factor that acts downstream from Notch during ovarian follicle cell development (*Sun and Deng, 2007*), was also expressed in developing and adult SCs but not epithelial cells (*Figure 3B–D*). Within the SUP lineage, secretory cells displayed the highest level of Notch activity and Hnt expression (*Figure 3D,D′*).

We extensively documented that Notch signaling and Hnt function during gland development using knockdown experiments (*Table 1*). Expression of *Notch*[RNAi] driven by *lzGal4* causes pupal lethality, however, flies in which Notch signaling is disrupted using dominant negative forms of the pathway components Psn or Su(H) survive to adulthood. When we examined the reproductive glands in females of these genotypes, no secretory cells were observed, the gland lumen was collapsed and the duct was malformed (*Figure 3E* and *Table 1*). Depletion of Hnt with two different *hnt*[RNAi] lines driven by *lz-Gal4* almost completely blocked secretory cell formation, while the gland lumen and duct developed normally (*Figure 3F,G* and *Table 1*). It would be worthwhile to further investigate the roles Notch signaling plays during specific steps in the secretory cell lineage. Because, these differential cell divisions (*Figure 3A*) probably resemble those extensively characterized during peripheral nervous system development, we initially focused on using this new information to generate glands containing reduced numbers of secretory cells, without disturbing the epithelial portion of the gland.

**Table 1.** The effect of altering secretory cell (SC) number in female reproductive glands on egg laying, ovulation, copulation, and sperm storage in spermathecae

| | Female glands | | Egg laying in 2 days | | Ovulation in 6 hr | | Copulation in 6 hr | | Sperm storage in 6 hr | |
|---|---|---|---|---|---|---|---|---|---|---|
| Genotype | N | SC/Female (Avg. ± SD) | N | Eggs/female/Day (Avg. ± SEM) | N | Ovulation (%) | N | Copulation (%) | N | Spermathecae with sperm (%) |
| *lzGal4* | 10 | 197.0 ± 18.0 | 45 | 38.9 ± 3.9 | 30 | 76.7 | 18 | 89.0 | | |
| *lz>cycA*[RNAi] | 36 | 2.0 ± 2.6* | 25 | 1.0 ± 1.0* | 30 | 3.3* | 23 | 56.5 | | |
| *lz>hr39*[RNAi] | 15 | 10.4 ± 7.4* | 25 | 5.2 ± 0.8* | 30 | 20* | 13 | 69.2 | | |
| *lz>hnt*[RNAi1] | 23 | 11.2 ± 8.4* | 45 | 8.0 ± 1.9* | 30 | 6.7* | 27 | 100.0 | | |
| *lz>hnt*[RNAi2] | 22 | 39.4 ± 12.1* | 25 | 16.5 ± 1.5† | | | | | | |
| *lz>Psn*[DN] | 25 | 1.8 ± 1.9* | 25 | 2.0 ± 1.8* | 25 | 4* | 25 | 100.0 | | |
| *lz>Su(H)*[DN] | 16 | 0.9 ± 1.0* | 15 | 2.5 ± 2.0* | | | | | | |
| *dpr5Gal4* | 5 | 192.0 ± 15.4 | 40 | 43.0 ± 4.5 | 31 | 64.5 | 25 | 100.0 | 50 | 98.0 |
| *dpr5>cycA*[RNAi] | 25 | 9.3 ± 3.7* | 25 | 3.3 ± 1.5* | 35 | 8.6* | 35 | 91.4 | 65 | 15.4* |
| *dpr5>hr39*[RNAi] | 10 | 191.3 ± 15.3 | 15 | 42.1 ± 2.6 | 24 | 45.8 | 21 | 90.5 | 37 | 83.8 |
| *dpr5>hnt*[RNAi1] | 18 | 17.1 ± 6.3* | 50 | 8.4 ± 1.8* | 34 | 2.9* | 34 | 94.1 | 64 | 9.4* |
| *dpr5>hnt*[RNAi2] | 13 | 99.5 ± 20.6* | 25 | 50.4 ± 2.8 | 24 | 79.2 | 24 | 95.8 | 49 | 81.6 |
| *dpr5>N*[RNAi] | 25 | 9.0 ± 3.2* | 25 | 0.9 ± 0.6* | 25 | 1* | 23 | 91.3 | 42 | 4.8* |
| *dpr5>Psn*[DN] | 16 | 83.9 ± 11.5* | 25 | 37.4 ± 7.5 | 26 | 46.2 | 26 | 100.0 | 52 | 88.5 |
| *dpr5>Su(H)*[DN] | 20 | 16.1 ± 4.5* | 25 | 26.2 ± 1.7† | 14 | 7.1* | 14 | 92.9 | 26 | 15.4* |

*p<0.001. T-test was used for secretory cell number and egg laying. Fisher's exact test was used for ovulation, copulation, and sperm storage.

†p<0.01.

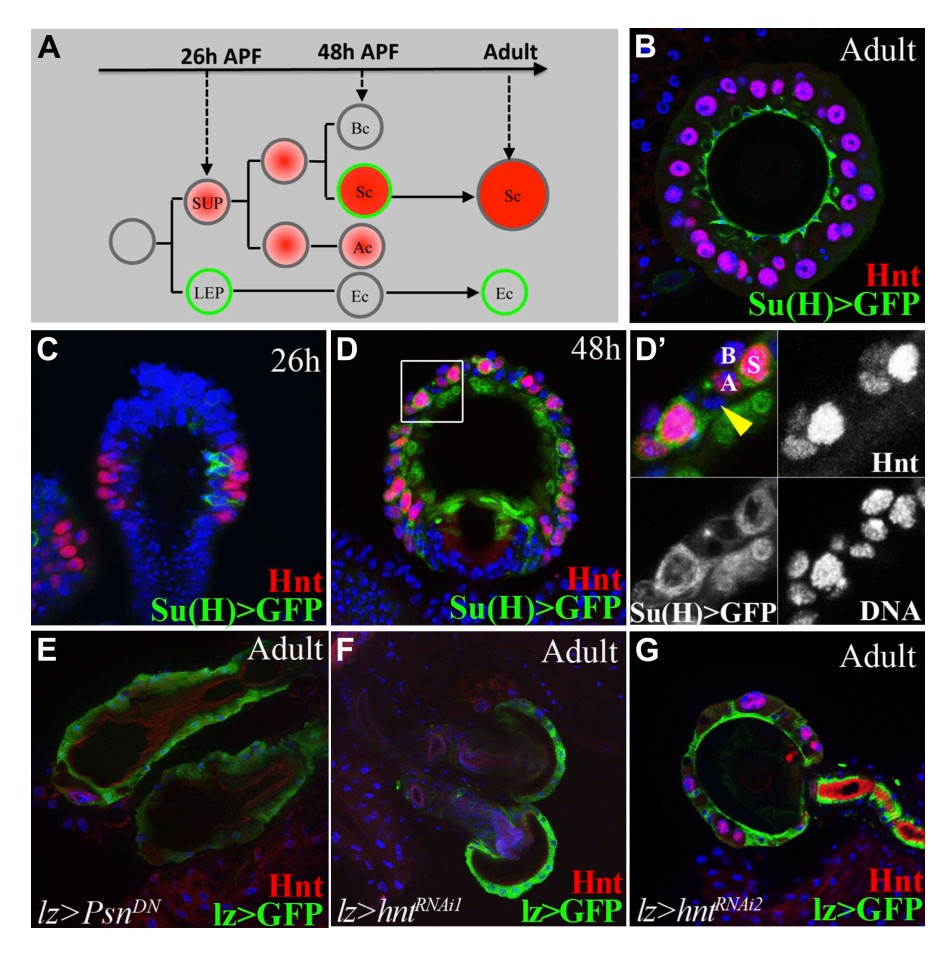

**Figure 3**. Notch signaling and Hindsight are required to form reproductive gland secretory cells. (**A**) The cell lineage underlying secretory cell development (**Sun and Spradling, 2012**). Notch signaling activity (green); Hnt expression (red). Ac: Apical cell; Bc: Basal cell; LEP: Lumen epithelial precursor; Sc: Secretory cell; SUP: Secretory unit precursor. (**B**)–(**D**) Notch activity (green) and Hnt (red) in spermathecae of adults (**B**), 26 hr pupae (APF) (**C**), and 48 hr APF (**D**). (**D'**) shows the boxed region from (**D**). Yellow arrowhead: Epithelial cell. (**E**)–(**G**) Adult spermathecae from females expressing lz>Psn[DN] (**E**) or lz>hnt[RNAi] (**F**–**G**) during gland development. lz (green) marks epithelial cells; Hnt (red) marks secretory cells.

## Female reproductive tract secretory cells regulate ovulation

Adult females whose reproductive glands are deficient in secretory cells were generated by knocking down *hnt* expression during pupal development using a *lzGal4* driver (**Figure 3F,G**), and tested for their ability to ovulate and lay eggs. SC-deficient females showed strong ovulation defects and laid significantly fewer eggs than controls (**Table 1**), indicating that secretory cells per se are required for ovulation. Females whose reproductive glands lack secretory cells were independently generated by expressing dominant negative (DN) forms of Psn (**Figure 3E**) or Su(H), and these females also had greatly reduced ovulation and laid few eggs (**Table 1**).

To further limit possible secondary defects present in animals that develop with reduced numbers of secretory cells, we searched for Gal4 drivers expressed specifically in female reproductive gland precursors among the Janelia Gal4 collection (**Pfeiffer et al., 2008**). From approximately 1000 lines screened, one Gal4 driver, 51B02 (termed *dpr5Gal4*) is specifically expressed in developing but not in mature secretory cells, nor in other reproductive tract or ovarian tissue (**Figure 4—figure supplement 1**). Using *dpr5Gal4* to drive a lineage marker confirmed its specificity for the secretory lineage of the reproductive tract and its absence in sensory neurons (**Figure 4—figure supplement 2**). Females with

different numbers of secretory cells were generated by knocking down *cycA*, *Hr39*, *hnt*, *N*, *Psn*, or *Su(H)* expression with *dpr5Gal4* (**Table 1**). Regardless of which gene was targeted, the ability of these females to ovulate and to lay eggs depended on the number of secretory cells in their reproductive glands (**Figure 4A,B**). Copulation was not affected (**Figure 4C**). These results demonstrate that one or

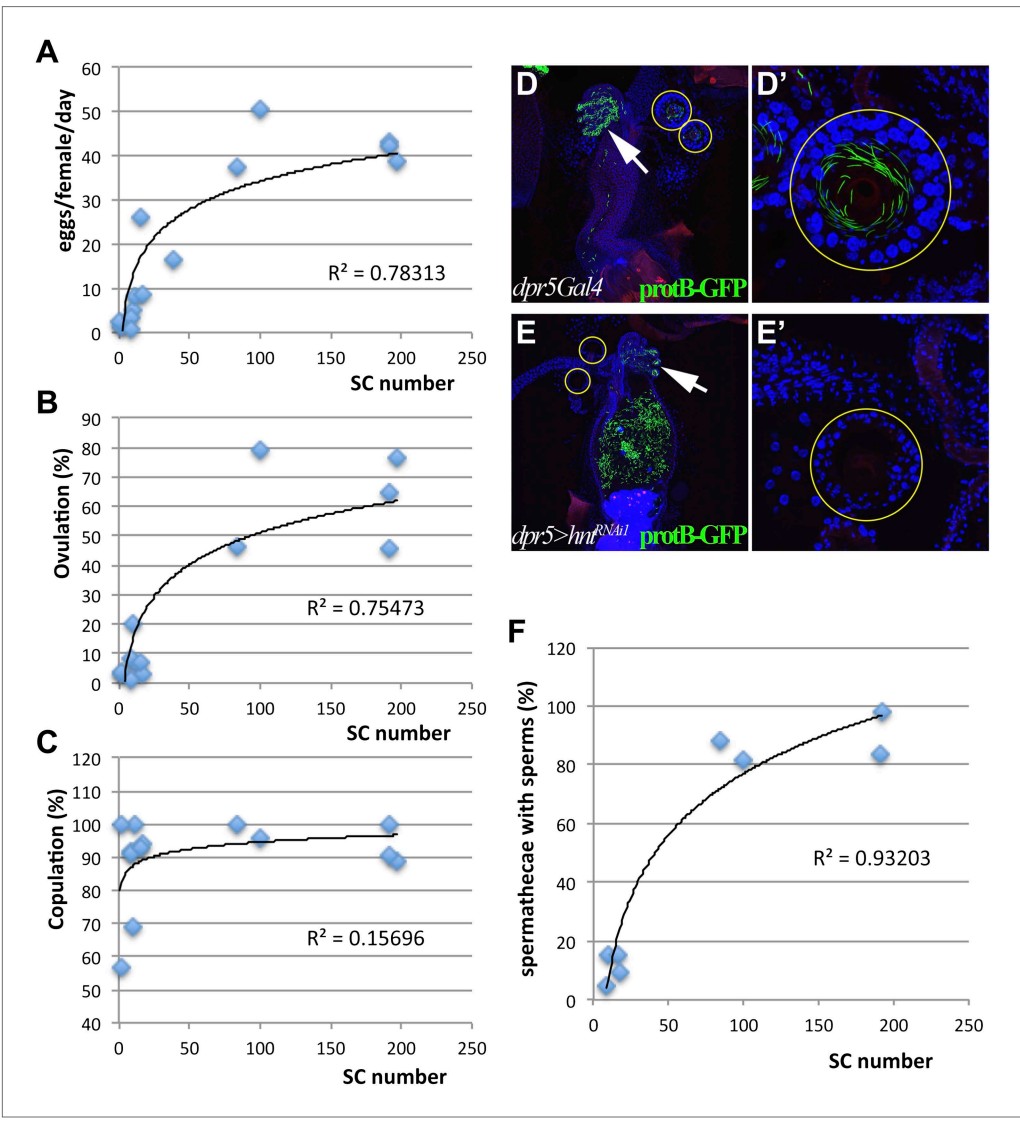

**Figure 4**. Female reproductive tract secretory cells mediate ovulation and sperm storage. (**A**)–(**C**) Relationship between secretory cell (SC) number and egg laying rate (**A**); percent ovulation (**B**); or percent copulation (**C**). Pooled data from genotypes in **Table 1**. Female reproductive tracts (**D** and **E**) and spermathecae (yellow circles in **D** and **E**; shown at higher magnification: **D'** and **E'**) from normal females (*dpr5Gal4* alone) (**D**) or females lacking SCs (*dpr5Gal4>hnt^RNAi*) (**E**) 6 hr after mating to males whose sperm nuclei are marked with protB-GFP (green). Seminal receptacle (white arrow). (**F**) Relationship between secretory cell number and the percentage of spermathecae with >5 sperm. Pooled data from genotypes in **Table 1**.

The following figure supplements are available for figure 4:

**Figure supplement 1**. The expression pattern of the *dpr5Gal4* line in spermathecae at 26 hr (using UAS-GFPnls), 39 hr APF (using UAS-GFP) and in the adult female lower reproductive tract (using UAS-GFP).

**Figure supplement 2**. Lineage-marked progeny of *dpr5+* cells (green) in the female reproductive tract, showing labeling of SC cells.

more products produced in the secretory cells of the reproductive tract are required for adult *Drosophila* females to ovulate and lay eggs.

## Female reproductive tract secretory cells are needed for sperm storage

Studies of animals lacking reproductive secretory organs (*Anderson, 1945*; *Allen and Spradling, 2008*) and of adults whose spermathecal secretory cells were partially ablated (*Schnakenberg et al., 2011*) have strongly argued that reproductive secretory cells produce products involved in sperm storage. We examined spermathecae for the presence of stored sperm in females generated as described above using *dpr5Gal4* that differ in secretory cell number. Within 6 hr after copulation, most wild type females had finished transferring sperm from the uterus to the storage organs (seminal receptacle and spermathecae) (*Figure 4D* and *Table 1*) (*Manier et al., 2010*). In contrast, less than 20% of females with a severe deficit of secretory cells (e.g., *dpr5Gal4>hnt^{RNAi1}*) had sperm inside the spermathaecal lumen at this time (*Figure 4E* and *Table 1*). Even among rare spermathecae that contained sperm inside the lumen, the number stored was much less than in controls. The absence of stored sperm is unlikely to be due to a physical block to the spermathecae, since sperm were found in the spermathecal duct and were stored in the seminal receptacle (*Figure 4E*). Our experiments showed that a minimum of about 80 secretory cells are needed for females to store a normal number of sperm (*Figure 4F*), indicating that a quantitative requirement exists for the products of these cells.

## Reproductive gland secretion and Hr39 action in adults are required for sperm storage

To further investigate the role of reproductive gland cells in adult female fertility, we sought to disrupt the activity of these cells during adulthood in glands that had developed normally. We shifted conditional mutations to the restrictive temperature only after eclosion (*Figure 5A*) and also used the Gal4 line *syt12Gal4*, which is expressed in mature secretory cells (*Figure 5B*), but not in the rest of the reproductive tract and ovary, to limit manipulations to adult secretory cells. The process of canonical protein secretion via the ER/Golgi/plasma membrane pathway was disrupted by expressing a dominant negative temperature sensitive allele of dynamin (*shi^{ts}*) (*Yang et al., 2009*), or by knocking down the *betaCOP* or *sec23* genes using RNAi controlled by the temperature sensitive Gal80 repressor (*Lee et al., 2004*; *Bard et al., 2006*; *Aikin et al., 2012*). When *syt12Gal4 UAS-shi^{ts}* adults were raised to the non-permissive temperature at eclosion, membrane trafficking in secretory cells was rapidly disrupted as expected (*Figure 5—figure supplement 1*). After 6 days at the non-permissive temperature, these females showed severe defects in sperm storage within the spermathecal lumen (*Figure 5B–D*), and those sperm that were stored exhibited an abnormal morphology characterized by a twisted sperm head (*Figure 5E–F*). Despite this, the females contained many sperm within the seminal receptacle (*Figure 5C*) and laid a near normal number of eggs (*Figure 5G*). Even stronger reductions in the number of sperm within the spermathecae were observed when *betaCOP* or *sec23* were knocked down in adults following the temperature shift to inactivate Gal80 (*Figure 5D*).

This experimental paradigm also revealed an ongoing requirement for *Hr39* in adults. When *Hr39* function was knocked down in adult secretory cells under the control of *Gal80^{ts}* (*Figure 5A*), severe reductions in the number of sperm stored in the spermathecae were also observed (*Figure 5D*). Those sperm that were present showed the same morphological defects seen in animals where canonical secretion had been reduced.

## Non-canonical protein secretion in reproductive secretory cells is required in adults for normal ovulation

The effects on ovulation of knocking down Hr39 expression or disrupting canonical protein secretion were particularly interesting. We modified our ovulation assay so that it could be applied not only to the initial oocytes, but to ongoing ovulation throughout several days of mature adulthood (see 'Materials and methods'). By determining the total number of eggs laid as well as the steady-state fraction of females that contained an egg in the uterus, we could calculate the average time oocytes spend during ovulation and within the uterus (*Table 2*).

Interestingly, females with ectopic adult expression of *shi^{ts}* in secretory cells were not defective in egg laying or ovulation (*Figure 5G,H*) despite their compromised ability to store sperm in the spermathecae. In particular, the time required per ovulation event was not increased compared to control (*Figure 5H*). Females in which canonical secretion was disrupted by knocking down *betaCOP* or *sec23* also laid eggs and ovulated similarly to controls (*Figure 5G,H*). Nonetheless, the secretory cell

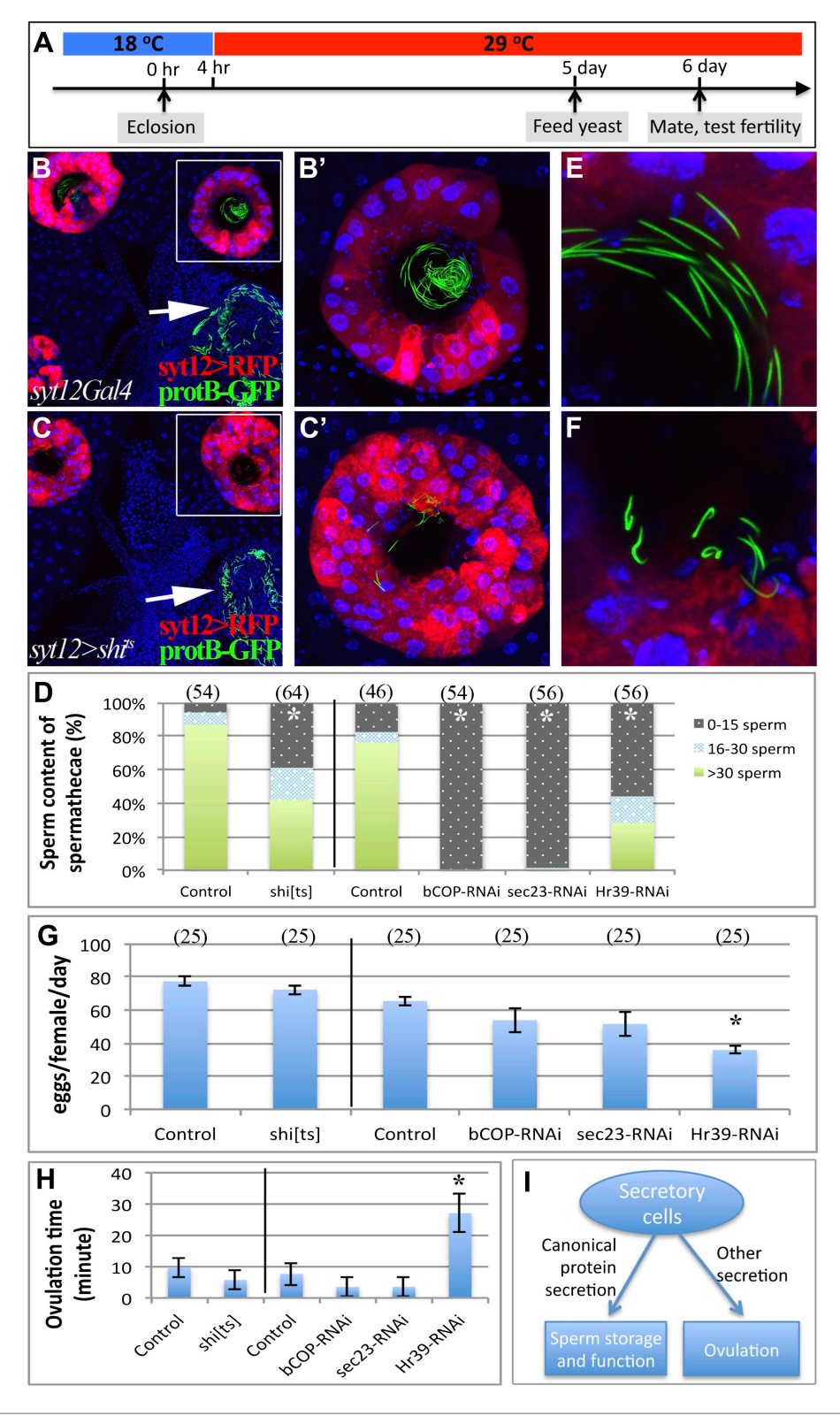

**Figure 5.** Canonical protein secretion from glandular secretory cells is required for sperm storage but not for ovulation. (**A**) Experimental scheme for testing adult secretory cell function using temperature sensitive *shi[ts]* or *GAL80[ts]*. (**B**) and (**C**) Dynamin (Shi) is required for sperm storage. Female reproductive tract of *syt12Gal4* control (**B**) or *syt12Gal4* driving *shi[ts]* expression (**C**) 6 hr after mating to protB-GFP males at 29°C. *syt12Gal4*
*Figure 5. Continued on next page*

*Figure 5. Continued*

expression is restricted to secretory cells as showed by UAS-RFP (red). (**B′**) and (**C′**): Higher magnification of boxed spermathecae; seminal receptacle contain sperm (white arrows). (**D**) Sperm content of spermathecae (three classes) is reduced in flies with indicated genotype (x axis) at 29°C. Bracket: Number analyzed. *p<0.01 (chi-square test). (**E**) and (**F**) Abnormal morphology of spermathecal sperm in *shi*[ts] females at 29°C (**F**) compared to control (**E**). Egg laying rate (**G**) and ovulation time (from ***Table 2***) (**H**) in flies with the indicated genotypes (x axes). *p<0.05 (Students t-test or Fisher's exact test). (**I**) Secretory cells use distinct secretory pathways to control sperm storage and ovulation.

The following figure supplements are available for figure 5:

**Figure supplement 1**. Membrane trafficking defects are observed in SCs when protein secretion is disrupted.

requirement that we had previously documented for the early ovulation was confirmed when we knocked down Hr39 in adult secretory cells. After six days at the non-permissive temperature, these animals showed a significantly lower rate of egg laying, about half of normal (***Figure 5G***) and ovulated much more slowly than controls (***Figure 5H***), requiring an average of 27 ± 6.1 min per egg compared to 7.6 ± 3.6 min in controls (***Table 2***). Our results show that at least two types of secretory cell products are released from reproductive gland secretory cells using different mechanisms. Products produced by the canonical protein secretory pathway are required to attract and store sperm in the spermathecae and to maintain their normal morphology. In addition, products needed to achieve a normal rate of ovulation require the function of *Hr39* in secretory cells, but do not utilize the canonical secretory pathway.

## Discussion

### *Drosophila* female reproductive tract secretions are required to attract and store sperm

Our experiments extend previous knowledge about the role reproductive tract secretions play in storing sperm. Sperm storage in the female reproductive tract is a general phenomenon in the animal kingdom including humans and insects (***Neubaum and Wolfner, 1999b***). In mammals, carbohydrate-dependent binding of sperm to the oviduct epithelia is important in order to form a sperm reservoir (***Talevi and Gualtieri, 2010***). In the absence of glands and hence of secretions, *Drosophila* sperm are still stored in the seminal receptacle, but they are poorly motile and fertility is low (***Anderson, 1945***; ***Allen and Spradling, 2008***). The same outcome is observed when spermathecal secretory cells are partially ablated in adults prior to mating (***Schnakenberg et al., 2011***).

**Table 2.** The effect of disrupting protein secretion or Hr39 expression during adulthood on the rate of egg laying and uterine egg content

| Genotype | N | Eggs/female/day | N | Uterus with egg (%) | Total time | Ovulation time | Uterus time |
|---|---|---|---|---|---|---|---|
| | | **Egg laying in 2 days*** | | **Egg distribution in 6 hr** | **Egg laying time (min)** | | |
| *syt12Gal4* | 25 | 77.3 ± 2.3 | 28 | 42.9 ± 18.3 | 17.1 ± 0.5 | 9.8 ± 3.1 | 7.3 ± 3.1 |
| *syt12>shi*[ts] | 25 | 72.2 ± 2.4 | 32 | 68.8 ± 16.1 | 18.3 ± 0.6 | 5.7 ± 2.9 | 12.6 ± 3.0 |
| *syt12Gal4* | 25 | 65.4 ± 2.3 | 29 | 62.1 ± 17.7 | 20.2 ± 0.7 | 7.6 ± 3.6 | 12.5 ± 3.6 |
| *syt12>βCOP*[RNAi] | 25 | 53.8 ± 6.3 | 28 | 85.7 ± 13.0 | 24.5 ± 2.9† | 3.5 ± 3.2 | 21.0 ± 4.0† |
| *syt12>sec23*[RNAi] | 25 | 51.5 ± 6.2 | 29 | 86.2 ± 12.6 | 25.7 ± 3.1† | 3.5 ± 3.3 | 22.1 ± 4.2† |
| *syt12>Hr39*[RNAi] | 25 | 35.8 ± 2.2† | 30 | 26.7 ± 15.8† | 36.9 ± 2.3† | 27 ± 6.1† | 9.8 ± 5.9 |

*1 day = 22 hr at 29°C.

†p<0.05. All data are mean ± 95% confidence interval. T-test was used for egg laying, while Fisher's exact test was used for egg distribution.

The work reported here allows several additional conclusions. First, the initial attraction of sperm to the spermathecae within 6 hr of mating requires a minimum amount of secretion from the secretory cells (SCs). Females with fewer than 25 SCs rarely contain sperm. In contrast, females with 80 or more SCs show the same high frequency of sperm in their spermathecae as wild type. The secreted attractive factor might interact with the accessory gland protein Acp36DE to facilitate uterine contraction (*Avila and Wolfner, 2009*), or act directly on sperm to regulate flagellar function (*Kottgen et al., 2011*; *Yang et al., 2011*). Second, the fact that sperm still move to the seminal receptacle in the absence of SCs shows that different mechanisms are involved in transporting sperm to the two different storage organs. Third, we found that female reproductive tract secretions are required to maintain sperm structurally. In the absence of secretory cells, sperm are not attracted to the spermathecae and those in the seminal receptacle aggregate and are difficult to individually assess (*Anderson, 1945*; *Allen and Spradling, 2008*). However, when protein secretion in SCs is disrupted using *shi*[ts], sperm that do make it to the spermathecae exhibit distinctive morphological abnormalities.

Finally, we documented that secretory cells competent to carry out canonical protein secretion and expressing *Hr39* are required to store sperm during adulthood. When protein secretion was disrupted after eclosion, sperm storage in the spermathecae was drastically compromised. Since we did not mate these females until the day they were tested, our experiments show that any initial accumulation of secretory products during pupal development turns over or is insufficient to store new sperm. The requirement for new secretion from the reproductive glands is consistent with previous studies showing that some proteins in these glands are induced by mating (*Mack et al. 2006*). The fact that Hr39 is required confirms that this gene, which is known to play a prominent role during reproductive gland development and to be expressed in adult secretory cells (*Allen and Spradling, 2008*), does play a key functional role in the adult gland. Many but not all spermathecal protein mRNAs, including many that are likely to be involved in sperm maintenance, are greatly reduced in an *Hr39* mutant that retains spermathecae (*Allen and Spradling, 2008*).

## *Drosophila* female reproductive tract secretions are required for ovulation

A major finding of this study is that the secretory cells of *Drosophila* female reproductive glands are required for efficient ovulation. When secretory cell number is deficient initially, ovulation is drastically reduced, despite normal copulation and the presence of sperm in the reproductive tract. When secretory cell function is reduced during adulthood by knocking down expression of *Hr39*, the time required for ovulation is greatly increased. Unlike the secretions that attract and stabilize sperm, ovulation is not disrupted by knocking down the protein secretory pathway. However, there are many possible reproductive tract secretions that might be released from the gland secretory cells by other mechanisms.

A previous study by Schnakenberg et al. (*Schnakenberg et al., 2011*) examined the role of spermathecal secretory cells by partially ablating them during adulthood using secretory protein regulatory elements to drive the apoptosis inducer Hid. Like these authors, we found that reproductive tract secretions are required to attract sperm to the spermathecae and to maintain their normal structure. We extended these observations by showing that a minimum number of about 80 secretory cells is required for normal sperm attraction and storage, and that these functions require the canonical protein secretory pathway and Hr39 expression. Schnakenberg et al. reported that egg release from the uterus frequently but sporadically is reduced in females with a deficit of secretory cells, and suggested that secretory cells produce an initial, long-lasting lubricant that coats the uterus. In contrast, when secretion was compromised, we saw that egg laying was strongly and consistently reduced due to defects in ovulation rather than in egg release. Schnakenberg et al. did not study ovulation independently from egg laying. In contrast, we used assays that separate these processes, allowing the role of secretion in ovulation to emerge.

## The control of ovulation may be conserved

In mammals, mature Graffian follicles compete for ovulation based on complex hormonal and biochemical signals that are closely tied to products locally produced by the follicle's granulosa and nascent luteal cells (*Mihm and Evans, 2008*). Much less is known about how individual follicles in the *Drosophila* ovary are selected for oviduct entry from a large pool. While nervous control of ovulation is clearly required to coordinate egg release with environmental and circadian factors (*Yang et al., 2008*), the underlying mechanism of egg selection is likely to be more complex and involve local

interactions as well as communication between the ovaries. Identification of the secretory cell product(s) that are required for ovulation would provide an important clue to uncovering these mechanisms.

A particularly attractive possibility is that communication signals between the reproductive tract and the ovary have been partially conserved between mammals and *Drosophila*. This prospect is now strengthened by the finding that Hr39 is required for ovulation, like its mammalian counterpart, Lrh-1. Prostaglandin-like molecules are known to regulate ovulation in mammals (*Dinchuk et al., 1995*; *Lim et al., 1997*), and Lrh-1 is thought to function by regulating expression of the prostaglandin-generating enzyme COX-II in mouse granulosa cells (*Duggavathi et al., 2008*). A prostaglandin-like signal already is known to function during egg maturation in *Drosophila* (*Tootle and Spradling, 2008*), and a COX-II-like enzyme, CG10211, is expressed in spermathecae under the control of Hr39 (*Allen and Spradling, 2008*). It will be interesting to determine if Lrh-1 functions in reproductive tract secretory cells. Finally, identifying additional glandular products acting in the *Drosophila* reproductive tract may elucidate additional pathways of communication between oviduct and ovary that are relevant to the induction of ovarian cancer.

# Materials and methods

## *Drosophila* genetics

Flies were reared on standard cornmeal-molasses food at 25°C unless otherwise indicated. Trans-heterozygous combinations of $Hr39^{7154/Ly92}$ (*Allen and Spradling, 2008*) and $lz^{3/34}$ (from Bloomington Drosophila Stock Center, BDSC, Bloomington, IN) were used for loss-of-function analysis. For the rescue experiment with *fruGal4* driving *UAS-mSP* (*Yang et al., 2009*), $Hr39^{7154}/Cyo$; *fruGal4* females were crossed to $Hr39^{7154}/Cyo$; *UAS-mSP* or $Hr39^{7154}/Cyo$. Knockdown with RNAi or dominant negative constructs were carried out at 29°C and the following lines were used: $UAS-cycA^{RNAi}$ (V32421; Vienna Drosophila RNAi Center, Vienna, Austria), $UAS-cdc2^{RNAi}$ (V41838), $UAS-Hr39^{RNAi}$ (V37694), $UAS-hnt^{RNAi1}$ (V101325), $UAS-hnt^{RNAi2}$ (V3788), $UAS-N^{RNAi}$ (gift from Sarah Bray), $UAS-Psn^{DN}$ (UAS-Psn.527.D447A, BDSC), $UAS-Su(H)^{DN}$ (*Mukherjee et al., 2011*), *UAS-mCD8:GFP, lzGal4* (BDSC), *dpr5Gal* (51B02; *Pfeiffer et al., 2008*), *ppkGal4* (*Yang et al., 2009*). In order to inhibit membrane recycling, the canonical exocytosis pathway, or Hr39 function in adult secretory cells, *syt12Gal4* (47E02; *Pfeiffer et al., 2008*) was crossed to $UAS-shi^{ts}$ (*Yang et al., 2009*), while *UAS-dcr2; syt12Gal4, tubGal80ts* was crossed to the RNAi line against bCOP (BDSC 33741), sec23 (BDSC 32365), or Hr39 (V37694) at 18°C. Virgin females were selected 4 hr after eclosion and immediately shifted to 29°C for 6 days. UAS-RFP was used to monitor *syt12Gal4* expression. For lineage tracing experiments, specific Gal4 driver was crossed to *G-Trace* lines to monitor real-time expression and lineage expression (*Evans et al., 2009*). Clonal labeling and pupae preparation were as previously described (*Sun and Spradling, 2012*). The Notch activity reporter *Su(H)GBE-Gal4, UAS-mCD8:GFP* was used to monitor Notch activation (*Zeng et al., 2010*), and ProtB-GFP was used to visualize sperm DNA (*Manier et al., 2010*). Control flies were derived from specific Gal4 driver crossed to wild-type Oregon-R.

## Egg laying, ovulation, and copulation tests

4- to 6-day-old virgin females were fed with wet yeast 1–2 days before egg laying experiments. Five females were mated to 10 Oregon-R males in each bottle covered with the molasses plate at 25°C and the number of eggs was counted every day for 2 days except for experiments that perturb the exocytosis pathway or Hr39 function in adult secretory cells, which were carried out at 29°C for 2 days. For ovulation and copulation tests, single-pair matings between a 4- to 6-day-old virgin female and a ProtB-GFP male were carried out in the morning at 25°C, except for experiments that perturb the exocytosis pathway or Hr39 function in adult secretory cells, which were carried out at 29°C. 6 hr after mating, females were dissected to examine eggs inside the reproductive tract, and the corresponding vials were also examined for laid eggs. Female reproductive tracts were then fixed with paraformadehyde and sperm inside them were examined to determine copulation success. The number of sperm inside spermathecae was manually counted. In *Table 2*, egg laying time (in minutes) = 22 × 60/number of eggs; the ovulation time = the egg laying time × (1 − egg distribution in uterus); and uterus time = the egg laying time × egg distribution in uterus. The 95% confidence intervals were calculated correspondingly.

## Immunostaining and microscopy

Pupal and adult reproductive tract staining was carried out as previously described (*Sun and Spradling, 2012*). Briefly, tissues were dissected in Grace's media, fixed in 4% EM Grade Paraformadehyde for 15–20 min, and blocked in PBTG (PBS + 0.3% Triton + 0.5% BSA + 2% normal goat serum). Incubation with primary antibody overnight was followed by a 2-hr incubation with secondary antibody and DAPI staining. Tissues were then mounted in Vectashield mounting media. The following primary antibodies were used: mouse anti-Hnt (1:75; Developmental Study Hybridoma Bank), rabbit anti-GFP (1:4000; Invitrogen), and chicken anti-β–Gal (1:1000; Abcam). Secondary antibodies were Alexa 488 and 546 goat anti-mouse, anti-rabbit, and anti-chicken (1:1000; Invitrogen). Images were acquired using the Leica TCS SP5 confocal microscope or the Zeiss Axioimager ZI microscope, and assembled using photoshop software.

## Acknowledgements

We are grateful to Dr Gerald Rubin for allowing us screen through the Janelia Gal4 collection. We also thank Drs Yuh Nung Jan, Sarah Bray, Utpal Banerjee, Scott Pitnick, and Steve Hou for sending us fly lines; Drs Chen-Ming Fan, Ming-Chia Lee, Vicki Losick, Matt Sieber, and Ethan Greenblatt for comments and discussion on the manuscript. ACS is an Investigator of the Howard Hughes Medical Institute.

## Additional information

### Funding

| Funder | Author |
| --- | --- |
| Howard Hughes Medical Institute | Allan C Spradling |

The funder had no role in study design, data collection and interpretation, or the decision to submit the work for publication.

### Author contributions

JS, Conception and design, Acquisition of data, Analysis and interpretation of data, Drafting or revising the article; ACS, Conception and design, Analysis and interpretation of data, Drafting or revising the article

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
