## [Decision Letter]

[Editors' note: this article was originally rejected after peer review. After an appeal against the decision and discussions between the authors and editors, the editors agreed that the submission could be published, subject to revisions. Both decision letters follow, along with the authors' response to the request for revisions.]

## Original decision letter including full reviews

Thank you for choosing to send your work entitled “Ovulation in *Drosophila* requires reproductive tract secretions controlled by the nuclear hormone receptor Hr39” for consideration at *eLife*. Your article has now been peer reviewed and we regret to inform you that your work will not be considered further for publication at this time. Your submission has been evaluated by 3 reviewers, one of whom is a member of our Board of Reviewing Editors, and the decision has been discussed further with one of eLife's Senior editors. The Reviewing editor and the outside reviewers discussed their comments before we reached this decision.

Each reviewer has acknowledged the importance of the work, but all three are unanimous in suggesting that the work presented is not complete and ready for publication. Reviewer 1 points to the lack of a detailed developmental analysis or identification of gene function, aspects that echo the comments of Reviewer 3. Similarly, Reviewer 2 feels that the nature of the secreted product, indeed the nature of secretion itself, is not fully examined, a view shared by Reviewer 1. Finally, Reviewer 3, an expert in the specific field, raises many technical questions and also would like to be further convinced that some of the points raised in different parts of the manuscript have indeed been settled by the results that are presented. The detailed comments of each reviewer are included below.

Reviewer #1:

The paper “Ovulation in *Drosophila* requires reproductive track secretions controlled by nuclear hormone receptor Hr39” explores an interesting link between secretions from female reproductive tract glands (Spermathecae and Parovaria) with ovulation in *Drosophila*. The authors show that a developmental defect in the specification of secretory organs in the female reproductive tract (through mutations in genes such as lozenge and Hr39) causes significant ovulation defects. These defects are neither due to mating deficiencies, nor caused by defects in neurons that innervate the oviduct muscles.

Based on the stereotypical division patterns seen in cells of the secretory unit precursor (SUP) cells and the expression of Notch signaling reporters at high levels, the authors test and establish that Notch signaling is required for the specification of secretory cells. Similarly, the authors establish a role for the transcription factor Hindsight, which is also expressed in these cells. Finally, using a Janelia Gal4 driver line expressed specifically in developing but not in mature secretory cells, the authors establish that secretory cells are needed in sufficient numbers to allow normal ovulation and sperm storage. Previous studies have shown that secretory cells in the female reproductive glands release their secretions into a structure called the end apparatus, which is in close apposition to the microvilli of the secretory cells. The authors test the role of secretions from these cells using a specific Gal4 and block exocytosis from these cells by over-expressing a dominant negative version of Dynamin (shi[ts]). This causes a defect in sperm storage but not in ovulation. The authors conclude that either the secretory factors from SC that regulate ovulation are independent of exocytosis or they are needed at very low levels, not disrupted in the chosen experimental design.

This is largely an observational paper, with few broader insights to report. That the secretory cells control ovulation, and that this process is Notch dependent is an interesting finding, described and proven with well-planned experiments and originality in the paper. But the manuscript does not explore much beyond this basic premise.

The data presented are of high standard but the paper lacks a cohesive narrative that moves the field forward. For example, the paper does not explore in detail the developmental aspects of the SC cells, and uses an assortment of mutations (in *Hr39*, *lz*, Hindsight, Notch) only as ways to eliminate the function of these cells rather than explore their interdependencies, or developmental relevance. Similarly, no attempt is made on the functional side to determine the nature of the secretions that have such profound effects on ovulation. In the end, the negative result with shibire-ts leaves the question wide open for further exploration. This work represents an exploratory project that has a lot of future potential but is not yet ready to move the field forward.

Reviewer #2:

The findings of Spradling and colleagues are potentially interesting. The paper clearly shows the requirement of secretory cells in the events leading to attraction and storage of the sperm. However, the data on the nature of the secretory components is not satisfactory. The authors report the involvement of Shibre (Dynamin); however, shibre is known to affect endocytosis in the neurons: the significance of this mutation on exocytosis is, therefore, unclear. It is important to know if the defect is due to secretion of extracellular matrix components or some growth factors or cytokines. I do not expect the authors to reveal the identity of the essential secreted component(s); they must, however, provide the clear involvement of the secretory process. Does it involve the regulated or the constitutive secretory pathway? The authors should test the effect of mutations in the genes that block constitutive (ER-Golgi-plasma membrane) and the regulated (Golgi-secretory storage granules-PM) in sperm attraction and attachment to support their proposal.

Reviewer #3:

There are a number of interesting conclusions in this paper; however, the title suggests a link between Hr39 and secretory cell function in the spermathecae, but the only clear role suggested by the experiments for Hr39 is a role in the formation of the spermathecae and the parovaria, not in the functioning of fully formed secretory cells in these tissues. In fact, using Hr39 RNAi with the dpr5-Gal4 driver showed what appears to be a small drop in ovulation after 6 hrs (from 64.5 to 45.8 although the RNAi control is missing) – however, no drop in egg laying after 48 hours was seen. In addition, no significant drop in sperm storage was seen. Based on these findings, it is difficult to see why the authors used Hr39 in the title of this paper. In addition it seems that the interesting conclusions are not highlighted in the Abstract or Introduction. There needs to be a clear differentiation in the writing of the paper between conclusions based on the absence of the entire tissue, and those effecting cell number on adult function. Overall, the paper should be rewritten to clearly highlight the more interesting conclusions regarding the role of the secretory cells of the spermathecae (i.e. ovulation defects and sperm storage defects), and the conclusions based on *hr39* and *lz* should be more clearly stated for what they are, a study of the loss of the tissues they help form.

Specific comments:

* The authors should state in the introduction the previously established link between the secretory cells of the spermathecae and sperm storage/ovulation – although Schnakenberg et al., (2011) is mentioned in the Discussion, it should also be mentioned up front in the Introduction.

* The authors need to clearly state that *lzGal4* expression has previously been shown to recapitulate actual *lz* expression. When does the *lzGal4* line come on in development?

* The authors state: “*lz* is expressed in the subset of these neurons near the oviduct-uterus junction, but the number of *fru*^*+*^*ppk*^*+*^ neurons was not affected by knocking down *cycA* or *hr39* using the *lzGal4* driver.” How do the authors know that knocking down *cycA* or *hr39* should change neuronal numbers? Do they know if *lz* is required for their proliferation? What if it has another function in these neurons?

* Why is *hr39* knockdown using RNAi tested in *lz* cells, but not *lz*-RNAi itself? Has *hr39* previously been shown to be expressed in these *lz* neurons? ppk-Gal4 with *hr39*-RNAi does not effect egg laying – why don't the authors test *lz*?

* The result in Figure 2E is *not* that mSP expression in *fru*^*+*^ neurons induces egg laying, an already published finding (Yang et al., 2009), but that mSP expression in *fru*^*+*^ neurons can not bypass the egg-laying defect in *hr39*^*-/-*^ mutants. Therefore, the SP behavioral switch can happen in the absence of *Hr39* (and therefore no spermathecae or parovaria are needed); however, the loss of egg-laying in *hr39*^-/-^ cannot be overcome by the SP behavioral switch.

* It appears from the methods that the shi[ts] flies are shifted just prior to mixing males and females. Is one possible explanation for the shi[ts] phenotype that all the important secretions for ovulation/egg-laying are already in place prior to mating?

* There appears to be no mention of statistical analysis: the types of analysis should be clearly stated along with every figure/table where they have been used and statistically significant differences/similarities should always be highlighted.

* *fru*^*+*^*/ppk*^*+*^ neurons have recently be shown to also be dsx-positive (Rezával et al., 2012).

* There are many different Gal4 lines used in this study: the authors should make it clear throughout the paper exactly where the lines have been shown to be expressed (either previously published or by the authors themselves). For example, the syt12-Gal4 line is introduced near the end of the paper but its overall expression pattern and reasons for use are far from clear.

* The first paragraph of the Discussion is confusing (for example, using double-negatives): this section should be re-worked to clearly state what was previously shown, and then clearly state what the current study adds to the story. The authors have shown a clear ovulation defect that was previously not shown; this is an interesting result that currently gets lost in the detailed explanation of the previous study.

## Decision letter outlining revision requirements

Thank you for choosing to send your work entitled “Ovulation in *Drosophila* requires reproductive tract secretions controlled by the nuclear hormone receptor Hr39” for consideration at *eLife*, and for telephonic discussions with Detlef Weigel and K. VijayRaghavan on December 21, 2012, which were useful. This has been followed up with discussions amongst the editors and we suggest the following points for a revision of the manuscript.

1)We strongly encourage you to change the title because it has been a source of confusion about what the paper is about. The current title can easily be read as implying that the paper will reveal the identity of the Hr39-dependent secretions. The title should instead reflect that this paper shows for the first time an unexpected way by which ovulation is controlled by the female reproductive tract.

2)The clear responses given to Referee 1 could be included in the text so that the paper better documents the assertion that this is indeed a foundational paper.

3)Next, experiments documenting the adult role of Hr39 should be included. Experiments that report the effects of disrupting at least one key gene in the ER/Golgi/PM secretory pathway(s) on ovulation would also greatly strengthen the manuscript. Your appeal letter indicates that both types of experiments are readily feasible.

4)Finally, a measured and brief analysis of your views on the Schnakenburg et al. paper, in the context of the current work, will be useful for readers to compare the two studies and to reach their own conclusions about the similarities and differences between them.

---

## [Author Response · Author letter]

Following the previous review, the editors made four requests for a revised version of our manuscript:

*1) We strongly encourage you to change the title because it has been a source of confusion about what the paper is about. The current title can easily be read as implying that the paper will reveal the identity of the Hr39-dependent secretions. The title should instead reflect that this paper shows for the first time an unexpected way by which ovulation is controlled by the female reproductive tract*.

We changed the title to “Ovulation in *Drosophila* is controlled by secretory cells of the female reproductive tract” to avoid any implications that specific secretions have been analyzed. “Hr39” was removed from the title as requested.

*2) The clear responses given to Referee 1 could be included in the text so that the paper better documents the assertion that this is indeed a foundational paper*.

We extensively rewrote the Abstract, changed a few lines in the Introduction, and modified the paragraph order in the Discussion to highlight the importance of the paper for studies of ovulation. A new section of Results and portions of the Discussion were modified to accommodate the new experiments. To further address why ovulation rather than Notch signaling should be a focus (as requested), we added a brief note in the section on Notch signaling.

*3) Next, experiments documenting the adult role of Hr39 should be included. Experiments that report the effects of disrupting at least one key gene in the ER/Golgi/PM secretory pathway(s) on ovulation would also greatly strengthen the manuscript. Your appeal letter indicates that both types of experiments are readily feasible*.

The main reason for the delay in resubmitting the revised version was to carry out both of these experiments. To unequivocally test the adult role of Hr39 (and the ER/Golgi/PM pathway) in adults, we drove RNAi expression targeting Hr39, betaCOP or Sec23 with Syt12-Gal4 in the presence of GAL80ts. At 18 **°**C the GAL80 inhibitor prevents Gal4 expression and the flies are phenotypically wild type with a normal number of secretory cells. We raised the temperature to 29**°**C at the time of adult eclosion. Some secretory products have probably been produced already at this time, and the turnover of the GAL80 protein at the restrictive temperature can take a few days, providing additional time for additional normal secretory cell function. Despite the potential for perdurance, when we tested the ability of these flies to store sperm, ovulate, or lay eggs at day 6 of adulthood, we saw strong defects (described below), thereby proving that Hr39 gene function and secretory cell function are required in adulthood. These results are presented in a revised Figure 5 and a new table: Table 2. The first two bars of panels D, G, and H present the original results, while the remaining bars show the new experiments. In addition, we added a new panel to Figure 5 with a timeline for the temperature shift and assay to make it clear that we are testing only adult function.

With respect to sperm storage, knocking down Hr39 in adulthood caused an extensive and statistically significant reduction in the ability of females to store sperm following mating (Figure 5D). When either betaCOP or sec23 was knocked down we observed even stronger effects on sperm storage than previously documented with shi[ts], in that all spermathecae now contain 15 or fewer stored sperm (see new Figure 5D). The few sperm that were present showed the same defective morphology illustrated in Figure 5F for shi[ts]. Moreover, knocking down betaCOP or sec23 by shifting the temperature beginning in 9-day-old adults (rather than at eclosion) and testing at 14 days of age gave the same result.

When we quantitatively analyzed egg laying and ovulation, these experiments also confirmed and extended our previous studies. We previously showed that shi[ts] did not cause a significant reduction in egg laying (over 24 hr) or in ovulation rate (measured in the first 6 hours after mating). We improved on these assays by measuring both the egg laying rate of the flies over 2 days beginning at day 6 (Table 2, column 1) and for a 6 hour sample of this period, measuring in individual flies of each genotype the number of eggs laid and the presence or absence of an egg still in the uterus (Table 2, column 2). (We confirmed that the percentage of flies with uterine eggs was the same after 6 hours and after the longer 2-day period.) By calculating the total time required for each laid egg, and by using the percentage of uterine eggs, we apportioned that time between ovulation time and time in the uterus (Table 2, column 3). Ovulation time provides a sensitive measurement of how efficiently ovulation is occurring in the different genotypes.

These results show that egg laying is not significantly reduced over 2 days by blocking secretion from reproductive secretory cells using shi[ts], betaCOP RNAi or sec23RNAi (Figure 5F), although it does trend downward, most likely due to secondary effects. Strikingly, ovulation time is not increased at all by disrupting ER/Golgi/PM secretion in these cells, confirming our previous result with shi[ts]. In contrast, Hr39-RNAi in secretory cells does significantly decrease the number of eggs laid and substantially and significantly slows ovulation (Fig. 5H). This confirms our conclusion that sperm storage requires ER/Golgi/PM secretion from female secretory cells and shows that this requirement exists in adulthood. Moreover, these data now show clearly that female secretory cells also function in adulthood to stimulate ovulation by a process requiring adult Hr39 expression, but that the stimulation of ovulation by secretory cells is independent of the canonical secretory pathway. We suspect that this second secretion is likely to be a small molecule that can move more easily than a large protein from the spermathecae and/or parovaria to the base of the ovary.

*4) Finally, a measured and brief analysis of your views on the Schnakenburg et al. paper, in the context of the current work, will be useful for readers to compare the two studies and to reach their own conclusions about the similarities and differences between them*.

We have added such comments to the Discussion as a second paragraph in the initial section on ovulation. Rather than discussing all the issues raised by their paper, we summarized the areas of agreement and the major difference: Schnakenburg et al. were unable to analyze ovulation whereas we discovered this connection between reproductive tract secretion, Hr39 expression, and ovulation.

Additional changes:

The Materials and methods and Figure legends have been revised to describe the new experiments. In addition, error bars that had been inadvertently omitted in the original version were restored, and additional statistical test information was added to Figure legends or the Materials and methods.